# Outcomes of Surgical Treatment for Graves’ Disease: A Single-Center Experience of 216 Cases

**DOI:** 10.3390/jcm12041308

**Published:** 2023-02-07

**Authors:** Hanxing Sun, Hui Tong, Xiaohui Shen, Haoji Gao, Jie Kuang, Xi Chen, Qinyu Li, Weihua Qiu, Zhuoran Liu, Jiqi Yan

**Affiliations:** Department of General Surgery, Ruijin Hospital, Shanghai Jiao Tong University School of Medicine, Shanghai 200025, China

**Keywords:** Graves’ disease, thyroidectomy, concomitant thyroid cancer, outcome, complication, intraoperative neural monitoring

## Abstract

Background: The role of surgery in the treatment of Graves’ disease (GD) needs to be revisited. The aims of the present retrospective study were to evaluate the outcomes of the current surgical strategy as a definitive treatment of GD at our center and to explore the clinical association between GD and thyroid cancer. Methods: A patient cohort of 216 cases from 2013 to 2020 was involved in this retrospective study. The data of the clinical characteristics and follow-up results were collected and analyzed. Results: There were 182 female and 34 male patients. The mean age was 43.9 ± 15.0 years old. The mean duration of GD reached 72.2 ± 92.7 months. Of the 216 cases, 211 had been treated with antithyroid drugs (ATDs) and hyperthyroidism had been completely controlled in 198 cases. A total (75%) or near-total (23.6%) thyroidectomy was performed. Intraoperative neural monitoring (IONM) was applied to 37 patients. The failure of ATD therapy (52.3%) was the most common surgical indication, followed by suspicion of a malignant nodule (45.8%). A total of 24 (11.1%) patients had hoarseness after the operation and 15 (6.9%) patients had transient vocal cord paralysis; 3 (1.4%) had this problem permanently. No bilateral RLN paralysis occurred. A total of 45 patients had hypoparathyroidism and 42 of them recovered within 6 months. Sex showed a correlation with hypoparathyroidism through a univariate analysis. A total of 2 (0.9%) patients underwent a reoperation because of hematomas. A total of 104 (48.1%) cases were diagnosed as thyroid cancer. In most cases (72.1%), the malignant nodules were microcarcinomas. A total of 38 patients had a central compartment node metastasis. A lateral lymph node metastasis occurred in 10 patients. Thyroid carcinomas were incidentally discovered in the specimens of 7 cases. The patients with concomitant thyroid cancer had a significant difference in body mass index, duration of GD, gland size, thyrotropin receptor antibodies and nodule(s) detected. Conclusion: Surgical treatments for GD were effective, with a relatively low incidence of complications at this high-volume center. Concomitant thyroid cancer is one of the most important surgical indications for GD patients. Careful ultrasonic screening is necessary to exclude the presence of malignancies and to determine the therapeutic plan.

## 1. Introduction

Graves’ disease (GD) is a syndrome characterized by an enlarged and overactive thyroid gland (Graves’ hyperthyroidism), which is caused by autoantibodies in the thyroid-stimulating hormone (TSH) receptor [1]. The conventional treatments for GD include antithyroid drugs (ATDs), radioactive iodine (RAI) therapy or a thyroidectomy; among them, a thyroidectomy is the definitive treatment. Recent studies have shown the limitations of ATD therapy based on long-term outcomes [2]; patients who received surgery as an initial treatment appeared to have lower chances of all-cause mortality, the lowest relapse and direct healthcare in the long-term when compared with those treated with ATDs or RAI [3]. It seems necessary to revisit the role of surgery in the treatment of hyperthyroidism.

Due to the risk of perioperative complications, surgery is suggested to patients usually after a careful evaluation. According to the literature, GD patients are at an increased risk of postoperative complications [4,5], including bleeding, vocal cord paralysis, tracheostomies and hypocalcemia. These risks appear to be lower at high-volume centers.

Moreover, a thyroid malignancy is one of the clinical situations that favor a thyroidectomy for Graves’ hyperthyroidism, as the American Thyroid Association recommends [6]. Concomitant thyroid cancer in GD is not rare. The correlation between GD and thyroid cancer is still not clear. The current thinking suggests that GD is not protective against thyroid carcinomas [7].

The aims of the present retrospective study were to evaluate the outcomes of the current surgical strategy as a definitive treatment of GD at our center and to explore the clinical association between GD and thyroid cancer.

## 2. Materials and Methods

The medical records of patients from 2013 to 2020 who had hyperthyroidism and underwent a thyroidectomy at this high-volume center—where over 1000 thyroidectomies are performed per year—were reviewed in detail. Only patients with hyperthyroidism caused by GD were involved in this study. Those who were lost to postoperative follow-up, with incomplete medical records or who had been previously treated with RAI were excluded. A total of 216 patients were eligible for the retrospective analysis (Figure 1). The patient characteristics, clinical data, surgical outcomes, postoperative complications and follow-up results were collected. The study was approved by the medical ethics review committee of Shanghai Ruijin Hospital.

The preoperative management of GD was performed by endocrinologists. The diagnosis was based on the clinical signs and symptoms of hyperthyroidism as well as thyroid hormones (T3, T4, free-T3, free-T4 and TSH) and the thyrotropin receptor antibody (TR-Ab). Among them, a rise in TR-Ab above the normal level was essential. All patients received a medication treatment with ATDs to achieve euthyroidism before surgery. A β-adrenergic blockade was prescribed if necessary. The majority of patients were euthyroid or mildly hyperthyroid at the time of operation, except the emergency one. A thyroidectomy was suggested to patients after a discussion with the multidisciplinary treatment team, which consisted of endocrinologists and surgeons. The indications for surgery were as follows: allergies or intolerances to ATDs; symptomatic compression or large goiters; a thyroid malignancy; coexisting hyperparathyroidism; and moderate-to-severe Graves’ orbitopathy (GO). Preoperative ultrasonography was routinely conducted. Thyroid nodules were evaluated according to the thyroid imaging reporting and data system (TI-RADS) grading. For nodules where there was a suspicion of malignancy (i.e., TI-RADS grade 4 or above), a fine-needle aspiration biopsy (FNAB) was suggested. However, a small proportion of patients who had suspicious nodules refused the FNAB for reasons of personal preference. An airway assessment was generally accomplished with a CT scan without contrast, which also favored the detection of vascular anatomical variations. A fiberoptic video laryngoscopy before surgery was always necessary in order to observe vocal cord motility. Not every patient had an iodine solution before the operation, which was only for those with glands enlarged beyond the outer edge of the sternocleidomastoid muscle, airway stenosis or a retrosternal goiter. For maximum benefit, Lugol’s iodine solution or a saturated solution of potassium iodide was taken continuously for 2 weeks prior to the surgery.

All the surgical procedures were carried out by the same experienced surgical team. The surgical extent could be divided into total and near-total thyroidectomies. However, in very few cases when an injury to the recurrent laryngeal nerve (RLN) was suspected, a sub-total thyroidectomy was adopted to avoid the further dissection of the nerves. An intraoperative rapid frozen section examination was routinely practiced. For patients who had nodules where there was a suspicion of malignancy without a FNAB, the intraoperative frozen section results directed the surgical extent. For those with concomitant thyroid cancer, a central compartment node dissection was added. A central compartment node dissection was also performed in case a malignancy could not be completely excluded in the light of frozen section results. A lateral cervical lymph node dissection was performed according to the FNAB result of the lymph node.

Intermittent intraoperative neural monitoring (IONM) was not applied until the second half of 2019 at our center. Due to the cost and medical insurance concerns, it was only permitted in these given cases: thyroid nodules where there was a suspicion of a malignancy with a diameter > 2 cm or airway stenosis caused by a large goiter. The patients suggested for IONM were fully informed and a signed consent form was required. To ensure successful monitoring, the placement of a recording surface electrode (NIM^®^ Standard Reinforced EMG Endotracheal Tube; Medtronic, Jacksonville, Florida, USA) should always be at the level of the vocal folds; this is difficult to achieve when severe retrosternal airway stenosis (shortest diameter of the trachea < 7 mm) exists. To exceed the lower edge of the narrow segment of the trachea, the endotracheal tube had to be placed at an excessive depth before the lobectomy with the aid of fiberoptic bronchoscopy. Once the unilateral lobe was pulled out from the retrosternal region, the endotracheal tube placement could be adjusted under a visual control by the anesthesiologist until the whole system was back to normal (Figure 2).

The tumor size (the largest dimension of the tumor) and the gland size (the largest dimension of the thyroid lobe) were measured on the specimen. An inadvertent parathyroidectomy was defined as the presence of the parathyroid gland or fragments in the specimen. All patients were followed up one month after their operation, then every three months until one year, then every six months until three years and more. The patients with hoarseness after the operation were examined with a fiberoptic video laryngoscope. In this study, those who had vocal cord paralysis over 6 months were thought to have a permanent RLN injury. Similarly, a parathyroid hormone (PTH) level below the normal limit with a calcium supplementation for more than 6 months after the operation was regarded as permanent hypoparathyroidism. Otherwise, the complications were recorded as transient. Most of the patients who had levothyroxine substitution therapy generally started one week after their operation.

For the descriptive statistics of the quantitative variables, the mean ± standard deviation (x ± s) was used. An independent sample *t*-test was used to compare the level of the quantitative variables between two groups if a homogeneity of variance existed. Otherwise, the Wilcoxon rank sum test was adopted. The chi-squared test was used to verify the correlation between the risk factors and postoperative hypoparathyroidism. The statistical analysis was performed with R (version 4.2.0). A *p* < 0.05 was considered to be statistically significant.

## 3. Results

The cohort of patients involved was composed of 182 females and 34 males. The mean age was 43.9 ± 15.0 years old. The mean body mass index (BMI) was 22.9 ± 3.3. The average duration of GD reached 72.2 ± 92.7 months. Most of them (211/216) had been treated with ATDs before surgery and only 23 patients had the iodine solution. Hyperthyroidism had been completely controlled (euthyroid or hypothyroid) in 198 cases. A total (75%) or near-total (23.6%) thyroidectomy was performed in most cases and 132 patients had a central compartment node dissection. A total of 104 (48.1%) cases were diagnosed as thyroid cancer; in 10 of them, a lateral cervical lymph node metastasis was present. IONM was applied only to a small number of patients (17.1%). The mean postoperative hospital stay was 3.02 ± 1.94 days. The surgery indications are listed in Table 1. Each patient may have had several indications. The failure of the ATD therapy was the most common (52.3%), followed by the suspicion of a malignant nodule (45.8%). The patients with moderate-to-severe GO came to surgery when euthyroidism was undesirably maintained under the medical therapy. In 13 (61.9%) cases, an improvement in GO was recorded. There was no worsening GO case after a thyroidectomy.

There were 24 (11.1%) patients who had hoarseness after the operation and a fiberoptic video laryngoscope was suggested (Table 2). In 15 (6.9%) of them, vocal cord paralysis was observed, but recovered within 6 months. However, 3 (1.4%) patients had this problem for more than 6 months after surgery; they were considered to have permanent RLN paralysis. No bilateral RLN paralysis occurred. In 2 cases with IONM during surgery, a loss of signal was recorded and vocal cord paralysis was subsequently confirmed; fortunately, these turned out to be transient. A total of 45 patients had hypoparathyroidism after a total or near-total thyroidectomy. Of these, 42 recovered within 6 months under a calcium replacement therapy, with the serum calcium and PTH levels back to normal. There were 2 (0.9%) patients who underwent a reoperation as a result of hematomas. No patient had a tracheal injury or a perioperative thyroid storm. Moreover, an inadvertent parathyroidectomy was found in 25 cases. Multiple factors were analyzed for postoperative hypoparathyroidism. However, only sex had a statistically significant correlation with this complication through the univariate analysis (Table 3).

During the follow-up, only one patient who underwent a sub-total thyroidectomy developed recurrent hyperthyroidism. RAI was then applied as a supplementary treatment to definitively control the hyperthyroidism.

Thyroid cancer was quite commonly found in the GD patients in this study. In most cases (72.1%), the malignant nodules were less than 1 cm in their maximum diameter (Table 4). Almost all the thyroid cancer was diagnosed as a papillary thyroid carcinoma. There was only one follicular carcinoma case. A central compartment node metastasis was present in 38 patients; however, 21 of them had less than 5 micrometastatic lymph nodes (<0.2 cm). A lateral lymph node metastasis occurred in 10 patients. Interestingly, there were seven (6.7%) cases in which thyroid carcinomas were incidentally discovered in the specimen (Figure 3). In three of them, according to the pathological results, the microcarcinoma was extremely small and could only be detected in the section of paraffin-embedded tissues. In the other four cases, the nodules were thought to be benign before the surgery.

To explore the clinical association between GD and thyroid cancer, multiple factors were analyzed between the sub-groups. In comparison with others, the patients with concomitant thyroid cancer had a significant difference in their BMI, duration of GD, gland size, preoperative TR-Ab and nodule(s) detected (Table 5). The existence of a malignant nodule interrupted the natural duration of GD, which could have been a chief reason to explain the smaller gland size, larger BMI and lower TR-Ab level in the thyroid cancer group.

## 4. Discussion

There are three treatment options for patients with GD. Although ATD therapy is the most common choice [8,9], a surgical option has the advantage of the immediate and definite treatment of hyperthyroidism without the toxicity associated with lifelong antithyroid medications or RAI [10]. However, GD patients are at an increased risk of postoperative complications compared with patients undergoing surgery for other indications [4,5]. The most common complications are hypoparathyroidism, laryngeal nerve palsy and hemorrhages.

According to the literature, the rates of postoperative hypoparathyroidism in GD patients range from 12.4–41.1% [4,5,11,12,13,14], which is a considerable variation. A lack of uniformity in the definition of complications could be one reason [10]. The use of prophylactic postoperative calcium supplements may decrease the risk of hypoparathyroidism [15]. The vast majority of hypocalcemia cases are transient. The rate of permanent hypocalcemia, which lasts for more than 6 months, is reported to be 0.5–3.8% [13,14]. In the present study, the rate of transient hypoparathyroidism was relatively low. Postoperative serum calcium with or without PTH within 24 h was measured in almost all cases to detect whether hypoparathyroidism existed. To prevent hypocalcemia symptoms, low-dose intravenous calcium supplements (calcium 1 g/d) were routinely given after surgery, based on the fact that a vitamin D insufficiency and an excess level of serum PTH were quite common before surgery at our center.

Postoperative hypoparathyroidism can be caused by various factors. In this study, after the univariate analysis, only the female sex proved to be a risk predictor. A former systematic review concluded that the perioperative PTH concentration and preoperative vitamin D level are useful biochemical predictors of post-thyroidectomy hypocalcemia. The clinical predictors of transient hypocalcemia include the female sex, Graves’ disease, parathyroid autotransplantation and an inadvertent parathyroid excision [16]. However, a recent large retrospective study indicated that age, BMI, sex, a high-volume center, previous bilateral thyroid/parathyroid surgery and parathyroid autotransplantation had a significant influence on postoperative hypoparathyroidism [17]. It should be noted that most of these studies were single-center cohort studies, with different definitions, assessments of outcomes and predictors involved.

An RLN injury is another important complication in terms of quality of life. However, a postoperative voice change is known to be a multifactorial issue; when dysphonia is regarded as a complication, the rate could be higher [18]. Those patients with large goiters were more likely to complain of voice changes, but several of them quickly recovered (within 48 h), which may have indicated a non-neural related voice issue or an early recovery RLN injury. To discern if an RLN injury existed, a fiberoptic video laryngoscope was routinely applied to observe the mobility of the vocal cords. The incidence of RLN injuries has been assessed using different methods in previous studies, which could be calculated per nerves at risk [13,17] or per patient [14]. In comparison, the rates of a transient RLN injury reported in the literature range from 3.5–11.5% in GD patients [13,14,17,19]; it is 0.2–2.3% for permanent injuries [12,14,19].

Whether IONM can reduce RLN injuries is still in dispute. For authors, it helps to identify the RLN and could be an important reference when a decision of a staged operation has to be made. To achieve accurate results, the standard application of monitoring techniques should be obeyed. In a few GD cases, the position of the endotracheal tube changes during the operation. An extreme laryngeal/thyroid retraction may occur during or after the delivery of a large sub-sternal or cervical goiter [20]. Concerns exist about a tracheomalacia with a postoperative tracheal collapse. An excessive depth of endotracheal tube placement seems to be reasonable. However, an adjustment to the EMG tube position with a fiberoptic bronchoscope or video laryngoscope for displacement during surgery is extremely important; this has been reported to occur in 5.7% of patients [21].

A thyroidectomy is thought to have no direct impact on the natural course of GO [22]. However, a thyroidectomy followed by a levothyroxine replacement therapy helps to rapidly control hyperthyroidism and reliably maintain the euthyroidism status, which are the main risk factors for GO progression. Earlier studies have reported higher regression rates (87.0–100%) of GO after surgery [23,24]. A recent randomized controlled trial found GO improved in only 36% of patients after a near-total thyroidectomy and 24% of patients after a total thyroidectomy within 12 months [13]. Different methods to define the course of GO and during the follow-up may explain the difference. In the present study, a relatively short follow-up may have influenced the final regression rate.

As ultrasonography may not be routinely conducted in GD patients and, as a portion of thyroid cancers were accidentally detected in specimens after surgery, it is hard to estimate the real rate of concomitant thyroid cancer in GD. Several large cohort studies have reviewed data in the area and found that thyroid cancer was detected in 1–1.7% [7,25,26] of GD patients during the follow-up. However, in surgical specimens, the event rate of thyroid carcinomas could be 7–55% [27,28,29,30,31,32] in GD. In the present study, the rate of thyroid cancer (48.1%) was relatively high in comparison. The ultrasound screening may have been an underlying cause. Most of the malignant nodules, especially the microcarcinomas (<1 cm in diameter), were detected by ultrasonography during the medication therapy.

Active surveillance, which has achieved favorable outcomes in several prospective clinical studies, is now regarded as an acceptable management strategy for low-risk papillary thyroid microcarcinomas (PTMCs) [33]. However, there is still a lack of reliable molecular markers or clinical features to differentiate unsuitable cases from the rest [34]. No studies investigating the influence of Graves’ disease on PTMCs under active surveillance have been reported so far. Several reports have drawn a similar conclusion that a low–normal TSH range may prevent carcinoma enlargements [35,36]. Paradoxically, the instability of the TSH level due to treatments for Graves’ disease may be inevitable during active surveillance.

In addition, the role that GD plays in thyroid cancer is controversial. A cohort study based on a Chinese population reported that GD patients were more likely to be subsequently diagnosed with cancer of the thyroid, with an adjusted hazard ratio of 10.4. An older GD cohort had a higher risk of developing cancer compared with a younger cohort. The risk of subsequent thyroid cancer for GD patients within six years was high [7]. However, a lower incidental papillary thyroid microcarcinoma rate was observed in GD patients compared with those with chronic lymphocytic thyroiditis, which may be explained by the immunologic conditions that could have a suppressive effect on cancer development in GD [37].

A meta-analysis showed that the presence of thyroid nodules was associated with a higher risk of thyroid cancer in surgically treated patients with GD; the number of nodules made no difference to the thyroid cancer risk [28]. A review of a surgical pathology archive noted that lymphocytic infiltration organized in germinal centers predicted not having differentiated thyroid cancers in GD patients [38]. In this study, multiple factors showed a significant difference between the sub-groups with or without thyroid cancer. However, the shorter duration of GD in patients with cancer could be responsible for the other three differences (BMI, gland maximum diameter and preoperative TR-Ab), given that the natural course of GD was cut short by surgery due to malignant nodules. As the failure of the ATD therapy was the most common surgical indication apart from cancer, GD patients may suffer larger goiters, a lower body weight and higher serum TR-Ab levels for longer without an early surgical intervention. Unsurprisingly, the incidence of thyroid nodules was significantly higher in the concomitant thyroid cancer group. This result was the same as reported in similar studies. The rate of chronic lymphocytic thyroiditis showed a rising trend in the cancer group, but no conclusion could be drawn. To understand the role in GD and thyroid cancer, a further pathological evaluation and comparison is needed.

Incidental thyroid carcinomas, which are not uncommon in GD patients, are generally uncovered with a thorough histological examination of the gland after surgery in a nodule or in thyroid tissue outside of the nodule. The incidence could be as high as 10.3%, according to a multicenter study [37]. In the present study, with routine ultrasonography screening, the rate of incidentally discovered thyroid carcinomas was lower than previously reported. Extremely small carcinomas and microcarcinomas in the context of large nodules seem to be the leading cause. Moreover, a minority of GD patients preferred to undergo the thyroidectomy directly without a FNAB when a suspicion of a malignant nodule existed. In these circumstances, it usually turned out to be malignant.

Whether Graves’ disease has an impact on the prognosis of differentiated thyroid cancer is also controversial. A multicenter retrospective study indicated that GD was associated with a worse outcome if the cancer was ≥1 cm [39]. On the contrary, an earlier study found that patients who underwent a thyroidectomy for Graves’ disease and who had a small thyroid cancer (<1 cm) had an excellent prognosis and longer disease-free survival than patients without GD [40]. In this study, the majority of malignant cases were microcarcinomas, but the rate of the lateral lymph node metastasis reached 9.6%, which was relatively high in comparison. It is hard to draw a conclusion as to whether differentiated thyroid cancer with GD is more aggressive. As the follow-up was too short to date, a further analysis will be carried out later.

## 5. Conclusions

A surgical treatment for GD was effective, with a relatively low incidence of complications at this high-volume center. In most cases, the postoperative hypoparathyroidism and hoarseness were transient. Concomitant thyroid cancer has become one of the most important surgical indications for GD patients. Careful ultrasonic screening is necessary to exclude the presence of malignancies and to determine the most appropriate therapeutic plan.

## Figures and Tables

**Figure 1 jcm-12-01308-f001:**
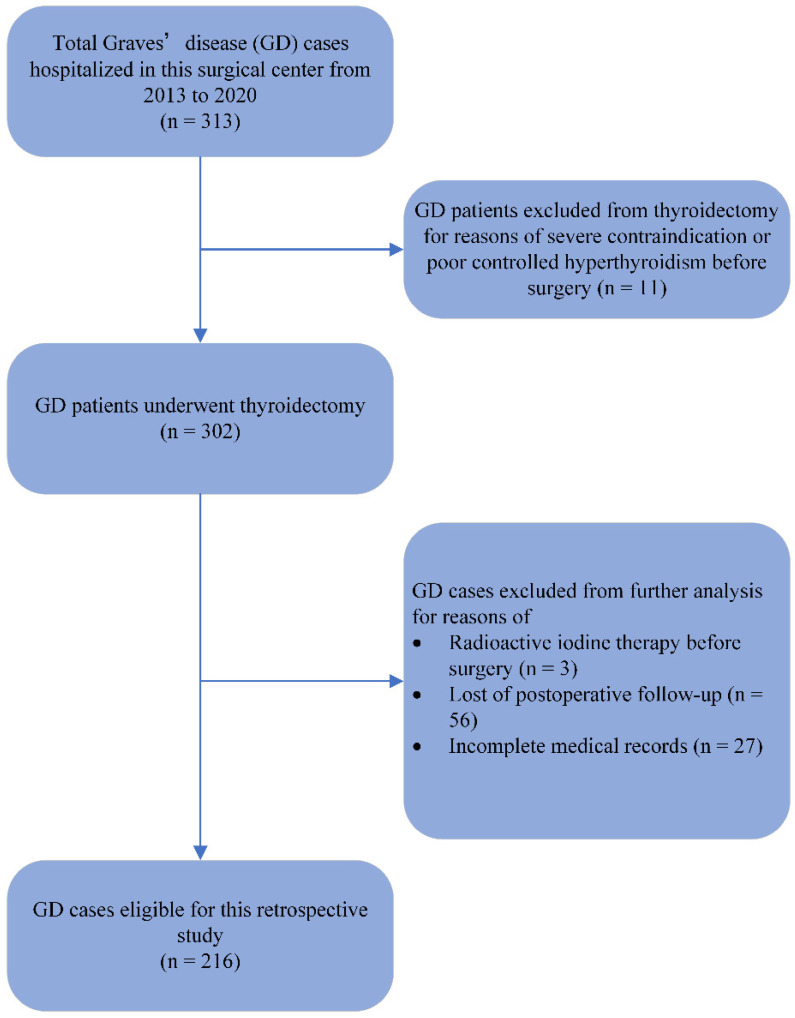
Flow chart of screening of eligible cases.

**Figure 2 jcm-12-01308-f002:**
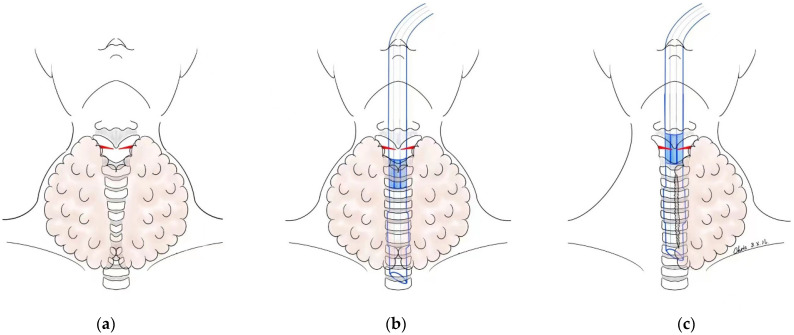
Adjustment of the EMG endotracheal tube position: (**a**). severe retrosternal airway stenosis; (**b**). the endotracheal tube had to be placed at an excessive depth before the lobectomy; (**c**). once the unilateral lobe was pulled out from the retrosternal region, the endotracheal tube placement could be adjusted.

**Figure 3 jcm-12-01308-f003:**
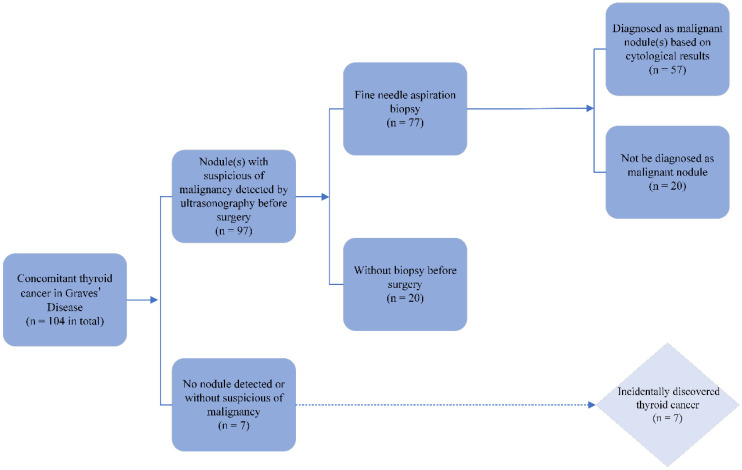
Nodule evaluation before surgery and incidentally discovered thyroid cancer.

**Table 1 jcm-12-01308-t001:** Basic characteristics of the 216 patients who underwent a thyroidectomy.

Characteristics	n (%)
Age (years)	43.9 ± 15.0
Sex	
Male	34 (15.7)
Female	182 (84.3)
BMI	22.9 ± 3.3
Duration of GD (months)	72.2 ± 92.7
Follow-up (months)	23.8 ± 20.3
Preoperation medication	
ATDs	211 (97.7)
Iodine	23 (10.6)
Thyroid hormone status	
Euthyroid	121 (56.0)
Hyperthyroid	18 (8.3)
Hypothyroid	77 (35.6)
Surgery extent	
Total thyroidectomy	162 (75.0)
Near-total thyroidectomy	51 (23.6)
Sub-total thyroidectomy	3 (1.4)
Central compartment node dissection	132 (61.1)
Lateral lymph node dissection	10 (4.6)
Intraoperation neural monitoring	37 (17.1)
Gland size (maximum diameter) (cm)	6.9 ± 2.5
Postoperative hospital stay (days)	3.02 ± 1.94
Pathological result	
Benign	112 (51.9)
Malignant	104 (48.1)
Surgery indications	
Failure of ATDs (allergy/intolerant/persistent or recurrent hyperthyroidism)	113 (52.3)
Suspicion of a malignant nodule	99 (45.8)
Local compressive symptoms	55 (25.5)
Coexisting hyperparathyroidism	1 (0.5)
Moderate-to-severe orbitopathy	21 (9.7)
Patient preference (for pregnancy or other reasons)	10 (4.6)
Retrosternal goiter	18 (8.3)

BMI: body mass index; GD: Graves’ disease; ATD: antithyroid drug.

**Table 2 jcm-12-01308-t002:** Complications after thyroidectomy.

Complications	n (%)
Hoarseness	24 (11.1)
Recurrent nerve paralysis	
Transient	15 (6.9)
Permanent	3 (1.4)
Bilateral recurrent nerve paralysis	0 (0)
Hypoparathyroidism	
Transient	42 (19.4)
Permanent	3 (1.4)
Hematoma (reoperation)	2 (0.9)
Tracheal injury	0 (0)
Thyroid storm	0 (0)

**Table 3 jcm-12-01308-t003:** Univariate analysis of risk factors associated with postoperative hypoparathyroidism.

Factors	Univariate Analysis	
	Odds Ratio (95% CI)	*p*-Value
Age (>50 years)	1.30 (0.55, 3.05)	0.552
Male	0.22 (0.06, 0.83)	0.017
BMI (>25)	0.72 (0.28, 1.88)	0.503
Benign/malignant	0.54 (0.24, 1.22)	0.137
Lymph node dissection	0.70 (0.30, 1.64)	0.409
Gland maximum diameter (>8 cm)	1.19 (0.51, 2.77)	0.695
Inadvertent parathyroidectomy	1.02 (0.31, 3.30)	0.975
Preoperative PTH (>70 pg/mL)	0.79 (0.33, 1.89)	0.593
Preoperative TR-Ab (>2 IU/L)	2.04 (0.69, 6.03)	0.192
Preoperative Vitamin D (<50 nmol/L)	1.46 (0.55, 3.85)	0.445

PTH: parathyroid hormone; TR-Ab: thyrotropin receptor antibody.

**Table 4 jcm-12-01308-t004:** Pathological results of concomitant thyroid cancer in GD patients.

Characteristics	n (%)
Tumor size (cm)	0.8 ± 0.6
Pathological diagnosis	
Papillary thyroid carcinoma	103 (99.0)
Follicular thyroid carcinoma	1 (1.0)
Other	0
Tumor focus	
Unifocal	76 (73.1)
Multifocal	28 (26.9)
T stage	
T1a	75 (72.1)
T1b	29 (27.9)
Other	0
Number of lymph nodes dissected	7.3 ± 8.6
Lymph node metastasis	
Central compartment node metastasis	38 (36.5)
Lateral lymph node metastasis	10 (9.6)
Incidentally discovered thyroid carcinomas	7 (6.7)

**Table 5 jcm-12-01308-t005:** Comparison between sub-groups of GD with or without thyroid cancer.

Factors	No Thyroid Cancer (*n* = 112)	Concomitant Thyroid Cancer (*n* = 104)	*p*-Value
Age (years)	44.63 ± 16.61	43.08 ± 13.08	0.703 ^†^
BMI	22.17 ± 3.27	23.57 ± 3.24	0.003 ^‡^
Duration of GD (months)	106.52 ± 106.96	35.51 ± 54.89	0.000 ^‡^
Gland maximum diameter (cm)	8.16 ± 2.60	5.57 ± 1.58	0.000 ^‡^
Preoperative TR-Ab (IU/L)	17.96 ± 15.81	6.95 ± 10.87	0.000 ^‡^
Gender (male/female)	18/94	16/88	0.890
Nodule(s) detected before surgery	63/112	101/104	0.009
Chronic lymphocytic thyroiditis	10/112	19/104	0.079

†: Independent sample *t*-test; ‡: Wilcoxon rank sum test.

## Data Availability

The data presented in this study are available on request from the corresponding author. The data are not publicly available due to privacy restriction.

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
