# Peer review of "Outcomes of Surgical Treatment for Graves’ Disease: A Single-Center Experience of 216 Cases"

_jcm, 2023, doi:10.3390/jcm12041308_

Round 1

Reviewer 1 Report (Previous Reviewer 3)

I have reviewed the new version and also find it acceptable.  The authors adequately addressed my comments.

Reviewer 2 Report (Previous Reviewer 2)

The revised manuscript is significantly improved.

This manuscript is a resubmission of an earlier submission. The following is a list of the peer review reports and author responses from that submission.

Round 1

Reviewer 1 Report

This paper reports the results of an analysis of the demographics and factors causing complications and hypocalcemia in 216 cases of Graves' disease at one institution, about half of which (104 cases) were complicated by differentiated thyroid cancer.

The frequency of postoperative recurrent nerve palsy, postoperative hypocalcemia, and postoperative hematoma is not significantly different from those reported in previous papers from other institutions, and there is nothing unique about the procedures performed at this institution. Nevertheless, the conclusion of this paper states that surgical treatment for Graves' disease is effective, with a relatively low incidence of complications. In fact, the results of this paper show that transient recurrent nerve paralysis was present in 6.9% of patients, permanent in 1.4%, transient hypoparathyroidism in 19.4%, and permanent in 1.4%. It is not clear what they mean by "relatively low incidence of complications.

In this paper, data on prognosis is not available because the cases covered in this paper are from 2013 to 2020. Therefore, it is not possible to evaluate the long-term prognosis of the treatment outcome.

Reviewer 2 Report

In the present retrospective single-arm cohort study, the authors evaluated the outcome of current surgical treatment for Graves’ disease (GD) and explored the clinical association between GD and differentiated thyroid cancer (DTC). Involving a cohort of 216 cases in a single center, and analyzing the data of patients’ characteristics, clinical information, surgical outcomes, postoperative complications, and follow-up results, the authors elucidated that surgical treatment for GD is effective and with a relatively low risk of complications. The authors indicated that differences in BMI, duration of GD, gland size, preoperative TR-Ab, and nodule(s) detected before surgery between subgroups of concomitant thyroid cancer and no thyroid cancer were statistically significant. However, the impact of GD on the prognosis of differentiated thyroid cancer remains controversial.

1) On Page 1, Lines 32-33, Conclusion. Concomitant thyroid cancer has become one of the most important surgical indications for GD patients. 

->The 2016 ATA guideline you cited recommends surgery. On the other hand, there is the concept of active surveillance. What do you think about active surveillance of low-risk papillary thyroid cancer in patients with controlled Graves' disease to avoid complications during surgery? Please add discussion.

2) On Page 2, Lines 82-83, the authors mentioned the nodules suspicious of malignancy without noting the criteria for judgment. Is it TI-RADS grade 3 and above?

3) In the 3. Results part, the number of cases with intermittent intraoperative neural monitoring (IONM) that developed recurrent laryngeal nerve injury is not listed separately. 

4) Both Figure 1 and Figure 2 are in low resolution. Please optimize them.

Reviewer 3 Report

The authors studied 216 patients who underwent a total or subtotal thyroidectomy for Graves Disease in a high volume thyroid surgery center.  They report the indications, the findings in regards to thyroid carcinoma, and complication rates, and make comparisons between groups with and without thyroid cancer as well as literature data.  The methods are nicely reported, and the results are straightforward.

The main problem with this report is that it is not unique.  Many institutions have reported results of thyroid surgery for Graves Disease, and this is a similar type of report.  The complication rates of hypoparathyroidism and recurrent laryngeal nerve injury are in the low-to-middle range of those reported by other institutions.

The high prevalence of thyroid cancer in the specimens primarily is due to selection bias, as patients with suspicious nodules on ultrasound or biopsy-proven papillary carcinoma were selected for thyroid surgery rather than being maintained on anti-thyroid drugs.

Other issues:

1.  The 216 patients were selected from a group of how many unique Graves Disease patients (i.e. the 216 is the numerator, what is the denominator)?

2.  What was the effect of thyroidectomy on Graves ophthalmopathy?

3.  In Table 1, you found 104 patients with malignancy, but 132 underwent central compartment dissection.  Why did 28 undergo an unnecessary central compartment dissection?

4.  You might provide the distribution of your patients with Graves Disease who underwent ADT alone, radioactive iodine, surgery alone, and surgery after ADT or radioactive iodine.  You should compare this distribution to other centers in other countries.